# The Open Question of How GPCRs Interact with GPCR Kinases (GRKs)

**DOI:** 10.3390/biom11030447

**Published:** 2021-03-17

**Authors:** M. Claire Cato, Yu-Chen Yen, Charnelle J. Francis, Kaely E. Elkins, Afzaal Shareef, Rachel Sterne-Marr, John J. G. Tesmer

**Affiliations:** 1Life Sciences Institute, University of Michigan, Ann Arbor, MI 48109, USA; catoc@usc.edu; 2Departments of Biological Sciences and Medicinal Chemistry and Molecular Pharmacology, Purdue University, West Lafayette, IN 47907, USA; yeny@purdue.edu; 3Biology Department, Siena College, Loudonville, NY 12211, USA; maple4lyfe64@gmail.com (C.J.F.); kelkins@nyit.edu (K.E.E.); sternemarr@siena.edu (R.S.-M.); 4Department of Chemistry and Biochemistry, Siena College, Loudonville, NY 12211, USA; ashareef@bu.edu

**Keywords:** G protein-coupled receptor, G protein-coupled receptor kinase, protein structure, phosphorylation, complexes, allostery, models

## Abstract

G protein-coupled receptors (GPCRs), which regulate a vast number of eukaryotic processes, are desensitized by various mechanisms but, most importantly, by the GPCR kinases (GRKs). Ever since GRKs were first identified, investigators have sought to determine which structural features of GRKs are used to select for the agonist-bound states of GPCRs and how this binding event in turn enhances GRK catalytic activity. Despite a wealth of molecular information from high-resolution crystal structures of GRKs, the mechanisms driving activation have remained elusive, in part because the GRK N-terminus and active site tether region, previously proposed to serve as a receptor docking site and to be key to kinase domain closure, are often disordered or adopt inconsistent conformations. However, two recent studies have implicated other regions of GRKs as being involved in direct interactions with active GPCRs. Atomic resolution structures of GPCR–GRK complexes would help refine these models but are, so far, lacking. Here, we assess three distinct models for how GRKs recognize activated GPCRs, discuss limitations in the approaches used to generate them, and then experimentally test a hypothetical GPCR interaction site in GRK2 suggested by the two newest models.

## 1. Introduction

G protein-coupled receptors (GPCRs) are the largest and the most diverse receptor family found in eukaryotes and are involved in regulating many diverse physiological processes. Upon activation by extracellular stimuli such as hormones and neurotransmitters, GPCRs transmit signals across the membrane to trigger intracellular signaling cascades. The hallmark seven-transmembrane topology of GPCRs was proposed in the 1980s based on the primary structure of bovine rhodopsin and the β-adrenergic receptor [1,2,3,4]. To date, more than 800 GPCR sequences have been identified in the human genome. Based on the sequence and functional similarity, the GPCR superfamily is divided into six classes, of which the rhodopsin-like family (group A) is the largest [5,6].

In cells, unliganded GPCRs exist in an equilibrium between inactive and active states, with inactive states being the most prevalent. Upon agonist binding, the equilibrium shifts toward active conformations that couple with downstream transducers such as heterotrimeric G proteins, GPCR kinases (GRKs), and arrestins. Heterotrimeric G-proteins are composed of Gα, Gβ, and Gγ subunits, with Gβ and Gγ forming a constitutive heterodimer (Gβγ). The activation of GPCRs induces conformational changes on the Gα subunit that allows the exchange of bound GDP for GTP. As a result, the Gα subunit dissociates from Gβγ subunits, and then the free Gα and Gβγ subunits regulate the activity of downstream effectors such as adenylyl cyclases, phospholipases, ion channels, and Rho guanine nucleotide exchange factors, thereby modulating diverse signaling pathways [7]. Meanwhile, GRKs phosphorylate the activated GPCRs, promoting the recruitment of arrestins to form GPCR-arrestin complexes, many of which interact with clathrin-coated pits for endocytosis and receptor internalization [8], and, ultimately, receptor downregulation [9]. Beyond their function in terminating G protein-mediated signaling from GPCRs, the binding of arrestins induces alternative G protein-independent signaling pathways, including the activation of ERK and Src family kinases [10]. 

The first atomic resolution GPCR structure determined was that of bovine rhodopsin bound to its intrinsic inverse agonist 11 cis-retinal [11]. Seven years later, the structure of the β2-adrenergic receptor (β2AR) in a complex with a diffusible inverse agonist was reported [12]. Both structures represented inactive, relatively stable conformations that revealed in atomic detail the basic core architecture for all GPCRs, wherein the extracellular N-terminus is followed by the characteristic seven transmembrane (TM) helices that are connected by three intracellular loops (ICLs) and three extracellular loops (ECLs). These elements are followed by a short amphipathic helix (H8) that packs against the inner leaflet of the plasma membrane and is often palmitoylated. Finally, the cytoplasmic C-terminus is typically an extended tail that contains multiple serine and threonine residues, some of which are phosphorylated by GRKs upon receptor activation. In some GPCRs, an extended third ICL (ICL3) can harbor the phosphorylation sites required for arrestin recruitment. 

The activated conformation of GPCRs is relatively unstable and difficult to isolate on its own. Therefore, many GPCR structures that have been resolved in an active state have employed protein complexes that recognize this conformation of the GPCR, such as G proteins or nanobodies [13,14,15,16]. The characteristic feature of an activated GPCR is a twisting movement of the cytoplasmic ends of TM5 and TM6 outward from the core of the TM bundle, forming a cytoplasmic cleft defined by residues in TM3, TM5, TM6, and the N-terminus of H8. This pocket accommodates the C-terminus of Gα subunits during G protein coupling. Recently determined structures reveal that arrestins also use a similar readout mechanism, but with an internal “finger loop” occupying the cytoplasmic pocket [17,18,19,20,21]. This provides a simple competitive mechanism for how arrestins block the access of G proteins to active receptors. However, arrestins also bind tightly to the phosphorylated C-tails of these receptors, and thus one arrestin molecule could, in principle, bind at one or the other or both sites, depending on the receptor structure, the extent of phosphorylation, and the phosphorylation pattern itself [22,23].

Ever since GRKs were identified as key regulators of active GPCRs, investigators have sought to determine which structural features on GRKs are used to discriminate between the agonist-free and agonist-bound states of GPCRs and how binding to agonist-bound GPCRs enhances the catalytic activity of the GRK [24,25,26,27,28]. It was also established early on that activated GPCRs are allosteric activators of GRKs, based in part upon the observation that activated receptors with no phosphosites can increase the GRK catalytic efficiency towards soluble peptide substrates [28,29], consistent with stabilization of a more catalytically competent state of the enzyme. However, despite a wealth of molecular information from high-resolution crystal structures of five of the seven human GRKs [30,31,32,33,34], the mechanism by which the kinase domain transitions from an inactive to an active state has remained elusive. This is because the N-terminus and active site tether (AST) regions of GRKs, which have long been postulated to play a role in the allosteric activation of GRKs, are typically disordered or adopt inconsistent conformations among the available structures, in part due to interactions mediated by crystal contacts. This behavior also reflects the fact that, with a few notable exceptions, the kinase domains in these structures do not adopt a fully closed conformation, at least based on a comparison to the transition state complex of protein kinase A (PKA) [35], a closely related and well-characterized member of the AGC kinase family to which GRKs belong (Figure 1A). The exceptions are a structure of GRK6 bound to an adenosine analog sangivamycin [36] and the recently reported structure of GRK5 in complex with Ca^2+^·calmodulin (Ca^2+^·CaM) [37]. In these models, the kinase domain, N-terminal helix, and AST adopt similar configurations (Figure 1B,C). In the first case, the assembly seems to be stabilized by interactions between the N-terminal helix with a fortuitous crystal contact and, in the latter case, by Ca^2+^·CaM. Thus, whatever the mechanism of GPCR-mediated GRK activation, it seems likely that it induces a kinase domain conformation similar to those exhibited by these two structures.

GRK activity is also dependent on its interactions with phospholipid bilayers. The seven GRKs are divided into three subfamilies and differ most in their C-terminal tails, which confer discrete mechanisms of membrane localization. The GRK1 subfamily members (GRK1 and 7) partition to membranes with the help of C-terminal farnesyl or geranylgeranyl modifications, respectively. The GRK4 subfamily members (GRK4, 5, and 6) contain N-terminal and C-terminal lipid-binding domains (NLBD and CLBD, respectively) that associate with anionic phospholipids in the plasma membrane [38,39,40,41]. GRK4 and GRK6 are also palmitoylated at their C-termini. The GRK2 subfamily members (GRK2 and 3) have a C-terminal pleckstrin homology (PH) domain that drives the membrane association via interactions with both anionic phospholipids and Gβγ [38], a process that also provides some allosteric feedback to the kinase domain [31,42]. The blockade of the interaction with Gβγ using a C-terminal fragment of GRK2 inhibits GRK2 activity in vitro and in cells [43,44,45,46]. Because the activity of GRK2 and GRK4 subfamily members are strongly dependent on the anionic phospholipids PIP_2_ and phosphatidylserine [47,48], it has been proposed that these also contribute to allosteric activation of the GRKs. It has been argued that such lipids use the same allosteric mechanism as activated GPCRs to stabilize the GRK in an active configuration [49].

Structural characterization at or near atomic resolution has been an important tool for developing mechanistic insights into how information is transferred in signal transduction pathways. As discussed above, such models are now available for various basal and activated GPCRs, GPCR–G protein complexes, and GPCR–arrestin complexes. Unfortunately, high-resolution data for GPCR–GRK complexes have not yet been reported, and modeling them remains highly dependent on the functional analyses and the available structures of the active GPCRs and (presumably active) GRKs (Figure 1B,C). There are currently at least three distinct models for the GPCR–GRK interaction. However, we currently lack both a critical analysis of these models based on their foundational data, and an extension of the unique regulatory aspects of the two newest models, which were based primarily from studies on individual GRKs, to the GRK family as a whole. Here, we attempt to address these gaps by assessing the methods used to generate these models and by testing whether a membrane proximal region of the regulator of the G protein signaling homology (RH) domain is a major GPCR-binding determinant in the GRK2/3 subfamily.

## 2. Materials and Methods

### 2.1. Materials

African green monkey kidney cells (COS-7) were purchased from the American Tissue Culture Collection (ATCC, Gaithersburg, MD, USA). N-terminal FLAG-tagged human β_2_AR cDNA in a mammalian cell expression vector (pcDNA3-FLAG-β_2_AR) and bovine GRK2, both wildtype (WT) and K220R, in a mammalian cell expression vector (pcDNA-GRK2) were provided by Dr. Jeffrey Benovic (Thomas Jefferson University, Philadelphia, PA, USA) [50,51]. pcDNA3.1-Gα_s_, pcDNA3.1-Gβ, and pcDNA3.1-Gγ (corresponding to the human proteins) were purchased from the Missouri University Science and Technology cDNA Resource Center. Isoproterenol and alprenolol were from Sigma, and peptide N-glycosidase F (PNGase) was from New England Biolabs. Polyclonal antibodies recognizing β_2_AR pSer355/pSer356, the β_2_AR carboxyl tail (to detect total receptor), and GRK2 were obtained from Santa Cruz Biotechnology. 

### 2.2. pcDNA-GRK2-H_6_ Preparation

The open reading frame of baculovirus vector pFBD-GRK2-H_6_ [52] was PCR-amplified using primers that appended the HindIII and EcoRI restriction sites and inserted into HindIII- and EcoRI-cut pcDNA3. The nucleotide sequence of the insert was determined by the Sanger method. The S670A variant was used because it eliminates an ERK phosphorylation site and thereby simplifies the purification of the homogeneous protein [53]. Site-directed mutagenesis was used to generate L33N, E36A, K220R, E532A, L536N, and L547N, and all constructs were verified by Sanger sequencing through the University of Michigan DNA Sequencing Core. Leucine was conservatively substituted with asparagine to specifically probe the importance of having a hydrophobic side chain rather than a difference in size. 

### 2.3. Protein Expression and Purification 

For the expression of GRK2 variants for kinetic studies, 0.4 L of suspension HEK293F cells purchased from Invitrogen were transiently transfected with 500 µg of the appropriate maxi-prepped pcDNA-GRK2-H_6_ at a polyethyleneimine-to-DNA ratio of 2:1 in Opti-MEM (Gibco). Cells were harvested by centrifugation at 3000 rcf 60 h post-transfection, flash-frozen in liquid nitrogen, and stored at −80 °C until future use. Frozen cell pellets were thawed and lysed in 50-mL CelLytic M (Millipore Sigma) per 1 L of expression volume for 30 min at 4 °C. After dilution to 60 mL in NiNTA Equilibration Buffer (25-mM HEPES, pH 8.0, 500-mM NaCl, 10-mM β-mercaptoethanol (β-ME), 10-µM leupeptin, and 100-µM phenylmethylsulfonyl fluoride (PMSF), the soluble fraction was isolated by ultracentrifugation at 40,000 rpm at 4 °C for 45 min. Clarified lysate was filtered through a 0.45-µm polyvinylidene fluoride filter. Filtered clarified lysates were passed over 0.5-mL NiNTA resin equilibrated in Equilibration Buffer by gravity. Resin was washed with 10 column volumes of High Salt Wash Buffer (25-mM HEPES, pH 8.0, 500-mM NaCl, 40-mM imidazole, 10-mM β-ME, 10-µM leupeptin, and 100-µM PMSF) and 10 column volumes of Low Salt Wash Buffer (25-mM HEPES, pH 8.0, 50-mM NaCl, 40-mM imidazole, 10-mM β-ME, 10-µM leupeptin, and 100-µM PMSF) prior to elution in 10 column volumes of Elution Buffer (25-mM HEPES, pH 8.0, 50-mM NaCl, 150-mM imidazole, and 10-mM β-ME). Eluted protein was diluted to 50 mL in Ion Exchange Buffer A (20-mM HEPES, pH 8.0, 25-mM NaCl, and 2-mM dithiothreitol (DTT) and loaded onto a tandem 1-mL HiTrap Q HP (GE Healthcare)/1-mL HiTrap SP HP (GE Healthcare) column setup. The tandem column setup was then disassembled, and GRK2 was eluted from the 1-mL HiTrap SP HP by a linear NaCl gradient prepared from Ion Exchange Buffers A and B (20-mM HEPES, pH 8.0, 1-M NaCl, and 2-mM DTT). Protein fractions that were >90% pure as assessed by SDS-PAGE and Coomassie staining were pooled, concentrated, and buffer-exchanged into Storage Buffer (20-mM HEPES, pH 8.0, 100-mM NaCl, 10% glycerol, and 2-mM DTT); flash-frozen in liquid nitrogen; and stored at −80 °C until use in steady-state assays. Typical GRK2 protein yields from HEK293F cells were 50–100-μg/L expression culture. 

### 2.4. Protein Concentration Normalization 

Prior to use in assays with purified GRK2, protein concentrations were normalized to the wild type as follows. Total protein concentration for each variant after a single freeze–thaw was determined by the Bradford analysis; then, 0.75 µg of each sample was separated on a 10% polyacrylamide gel by SDS-PAGE, and the gel was stained using Bio-Safe Coomassie Stain (Bio-Rad). Band intensities corresponding to GRK2 were analyzed in ImageQuant. The wild-type GRK2 concentration determined by the Bradford analysis was then used as a reference to adjust the concentrations of the other variants based on their relative band intensities. 

### 2.5. Determination of Steady-State Parameters 

Steady-state parameters for the phosphorylation of tubulin and rhodopsin in rod outer segments (ROS) by GRK2 with variable ATP concentrations were determined at room temperature in the Reaction Buffer (20-mM HEPES, pH 7.5, 2-mM NaCl, 2-mM MgCl_2_, and 2-mM DTT), and the reactions were stopped at 8 min within the linear range of the GRK2 kinase reaction for both tubulin and rhodopsin (data not shown). Steady-state parameters for the phosphorylation of rhodopsin in ROS were also determined with variable rhodopsin concentrations. For variable rhodopsin experiments, reactions containing 20-nM GRK2 and 150-nM human Gβ_1_γ_2_ in the Reaction Buffer were incubated with 50-nM–5-μM light-activated rhodopsin in ROS and initiated by the addition of 5-μM ATP supplemented with radioactive [γ-^32^P] ATP. Reactions were quenched at 8 min with SDS gel loading buffer, and phosphorylated products were separated on a 4–15% Criterion TGX precast gel. Gels were exposed to a phosphor screen overnight, scanned using a Typhoon scanner, and band intensities corresponding to phosphorylated product were quantified using ImageQuant software. For each independent experiment, band intensities were normalized to the wild type, such that the highest wild-type band intensity was set to 1 to account for the day-to-day variability in the phosphor screen intensity. Each experiment was performed in triplicate. *K*_m_ and normalized V_max_ were determined by plotting the normalized band intensity as a function of either ATP or rhodopsin concentration, and fitting it to the Michaelis–Menten equation. Statistical significance was assessed by one-way ANOVA using multiple comparison corrections. All curve-fitting and statistical analyses were performed using GraphPad Prism 7.03.

### 2.6. Agonist Dose- and GRK2-Dependent Phosphorylation of β_2_AR in Cells

COS-7 cells were grown in Dulbecco’s Modified Eagles Medium (DMEM) GlutaMAX (ThermoFisher Scientific, Waltham, MA, USA) supplemented with fetal bovine serum (10%) and penicillin/streptomycin/amphotericin B (Fungizone, 100 units/mL) at 37 °C with 5% CO_2_. Cells (3 × 10^5^ cells/well) were plated in 6-well tissue culture dishes and transfected the following day with 300-ng pcDNA3.1-FLAG-β_2_AR and 400 ng each of pcDNA3.1-Gα_s_, pcDNA3.1-Gβ, and pcDNA3.1-Gγ and, when present, 400-ng pcDNA3-GRK2 using 6- to 7-µL FuGENE-HD. The total amount of DNA was brought to 2000 ng with salmon sperm DNA. For the isoproterenol (ISO) dose-response experiments, 40–44 h after transfection, cells were serum-starved for 1 h before a 5-min treatment with a range of isoproterenol (ISO) concentrations (100 pM–10 µM) or 10-µM ALP as the “zero” concentration. For the assay of β_2_AR phosphorylation by GRK2-H_6_-RH domain mutants, cells were transfected as described above, except WT and mutant pcDNA-GRK2-H_6_ replaced the pcDNA-GRK2. After 40–44 h, cells were serum starved for 1 h and then treated for 5 min with 2-nM ISO. Cells were washed twice with cold 20-mM Tris, pH 7.5 and 150-mM NaCl and scraped in 200-µL/well receptor solubilization buffer (20-mM HEPES, pH 7.4, 150-mM NaCl, 10-mg/mL dodecylmaltoside, 10-mM DTT, 1-mM PMSF, 10-µg/mL leupeptin, 200-µg/mL benzamidine, 20-mM tetrasodium pyrophosphate, and 10-mM NaF). The resulting lysates were solubilized by mixing for 30 min on an orbital shaker at 4 °C and clarified by centrifugation at 16,000× *g* at 4 °C. Soluble fractions were treated with PNGase (3.3 U/uL) for 2 h at 37 °C and immunoblotted with (a) β_2_AR phosphosite antibody that recognizes agonist-induced phosphorylation (pSer355/pSer356), (b) an antibody that recognizes the β_2_AR carboxyl tail and reflects total β_2_AR, and (c) GRK2 polyclonal antibody, which were carried out as described [52,54]. Each primary antibody was visualized with horseradish peroxidase-conjugated goat anti-rabbit secondary antibody (Bio-Rad, Richmond, CA, USA) and chemiluminescent substrates (ThermoFisher Scientific, Waltham, MA, USA), either SuperSignal West Femto for pSer blots or SuperSignal West Pico for β_2_AR and GRK2 immunoblots. Signals were visualized using the Bio-Rad ChemiDoc XRS System, and band intensities were quantified using Bio-Rad Quantity One software. GraphPad Prism v.6 (La Jolla, CA, USA) was used for nonlinear regression curve fitting and statistical analysis. To calculate EC_50_ in the ISO dose-response assay, data were scaled to 100 for the entire dataset, and values obtained with and without GRK2 were fit to a sigmoidal curve (four-parameter logistic equation) with a variable slope. For the assay of the RH domain mutants, the transfected GRK2-dependent signal was normalized for minor variations in the amount of GRK2 and receptor in the cell lysates and scaled to 100 with the GRK2-H_6_ WT value. Repeated measures one-way ANOVA and Dunnett’s multiple comparisons test were used to compare the statistical significance of differences between WT GRK2-H_6_ and the mutant derivatives.

## 3. Results

### 3.1. Analysis of GPCR–GRK Interaction Models

We first assessed three leading GPCR–GRK interaction models in light of the approaches used to derive them and their unresolved issues.

#### 3.1.1. The N-Terminal Helix as the Primary GPCR Docking Site (Model 1)

Like PKA, the GRK kinase domain consists of small and large lobes. The ATP-binding site is found in a deep crevice formed at their interface, whereas peptide substrates are presumed to bind to the large lobe. They feature an AST loop [55] that, in PKA and other AGC kinases, is disordered until the kinase adopts an active configuration. Unlike PKA, GRKs do not require phosphorylation of their so-called activation loops to help them assemble into an active form, and they contain accessory structural elements, including the RH domain [56] and a ~20 amino acid N-terminal helical region known to be essential for receptor phosphorylation [36,54,57,58,59,60,61,62]. 

The crystal structure of the GRK6–sangivamycin complex [36] was the first to exhibit a kinase domain conformation similar to that of active PKA and the first to feature a fully ordered N-terminus and AST loop. The structure was thus hypothesized to represent the conformation of a GRK when bound to an activated GPCR (Figure 1B). In this structure, the N-terminal 17 amino acids form a single α-helix that packs against both the small lobe and the ordered AST loop, thereby forming a stabilizing bridge between the small and large lobes over the active site and near the hinge of the kinase domain. Based on the hypothesis that the receptor binds primarily to the N-terminal helix and that this drives allosteric activation of the GRK (in this case kinase domain closure), extensive structure-guided mutational analyses of GRKs from each of the three GRK subfamilies was carried out to identify residues in the N-terminal helix [36,54,62], as well as the AST region and nearby regions of the small lobe [61,63] that impair receptor phosphorylation but not soluble peptide phosphorylation. To interpret the results, it was assumed that mutations affecting both receptor and soluble peptide phosphorylation would specify residues that either stabilize the activated kinase conformation or are directly involved in catalysis, whereas mutations that primarily impair receptor phosphorylation would specify residues that either make direct interactions with the receptor or are involved in receptor-mediated allosteric activation. The results from each GRK subfamily consistently implied that residues in the N-terminal helix that faced away from the core of the kinase domain solely impaired GPCR phosphorylation, whereas those that formed contacts with the small lobe or AST were important for both receptor and soluble peptide phosphorylation. Thus, a region comprised of the N-terminal helix, AST-loop, and surrounding surfaces on the small lobe of the kinase domain small lobe likely form complementary interactions with the cytoplasmic surface of activated receptors, which would in turn stabilize the active state of the GRK. The most speculative aspect of this model is that the extreme N-terminus of the N-terminal helix is docked directly into the cytoplasmic cleft that forms in activated GPCRs, a mode of interaction analogous (although in opposite polarity) to the binding of the C-terminal helix of heterotrimeric G protein α subunits [36] (Figure 2A).

This model however lacks direct structural evidence. Regardless, in the recently reported structure of the GRK5–Ca^2+^·CaM complex, the kinase domain adopts a conformation similar to that of the GRK6–sangivamycin complex [36], including the N-terminal helix and AST region [37] (Figure 1C). In this structure, the N-terminal domain of Ca^2+^·CaM binds to the kinase domain in a manner similar to that proposed for GPCRs in this model (Figure 2A), supporting the idea that a protein–protein interaction is required to stabilize the network of interactions formed between the N-terminal helix, the small lobe, and the AST in activated GRKs.

#### 3.1.2. The NLBD as the Primary GPCR Docking Site (Model 2)

Like the N-terminal helix, the GRK RH domain is not found in other AGC kinases, making it an equally attractive hypothetical GPCR docking site. Moreover, its terminal and bundle subdomains engage the small and large lobes of the kinase domain, respectively, in a way highly reminiscent of SH3 and SH2 domains of Src [30]. Thus, a model for receptor activation would be to disrupt these interdomain contacts and thereby allow the kinase domain to relax into an active state. Some experimental support for this line of reasoning initially came from studies that showed the RH domain of GRK2 could bind directly to the metabotropic glutamate receptor [65]. Recently, a β_2_AR-GRK5·sangivamycin assembly was isolated in PIP_2_-containing bicelles and used to investigate their interface via crosslinking with mass spectrometry (CLMS), electron microscopy (EM), hydrogen-deuterium exchange mass spectrometry (HDX-MS), and molecular dynamics [49]. Although these assemblies exhibited kinase activity, a caveat is that the particles could be formed in the presence of either agonist (BI-167107) or inverse agonist (ICI-118551), making it unclear whether there was formation of an agonist-dependent GRK complex. This study also took advantage of GRK5 mutations that both strengthen and weaken an “ionic lock” (not to be confused with the ionic lock of GPCRs [11,66]) proposed to form between the RH bundle subdomain and the large lobe of the kinase domain (Figure 1A). Breaking this interface with disruptive mutations in the RH bundle or large lobe of GRK5 resulted in a mild 1.6–2.5-fold enhancement of kinase activity toward activated β_2_ARs [49,67]. However, similar mutations in GRK2 [42] and GRK1 [64] were not activating, and the activation in GRK5 does not require a mutation of any ionic side chains [67]. In 2D negative-stain EM class averages [49], the GRK5 variant with a disrupted RH-large lobe interface displayed more open and elongated conformations, suggesting that disruption allows increased kinase domain flexibility in GRK5. In contrast, GRK5 variants where the interface was stabilized by an engineered disulfide bridge lost kinase activity and exhibited more compact conformations. Wild-type GRK5 was, however, not compared as a control in this experiment to establish whether a dynamic RH bundle–large lobe interface is unique to the more active interface mutants. 

Bissulfosuccinimidyl suberate (BS3) crosslinking of the β_2_AR-GRK5 mixture, followed by tandem mass spectrometry, suggested three primary interfaces: ICL3 of the β_2_AR with the NLBD and CLBD of GRK5, ICL2 of the β2AR with the RH bundle subdomain of GRK5, and the C-terminus of the β_2_AR with the GRK5 kinase domain [49]. The observed crosslinking between ICL3 of the β_2_AR and the lipid-binding domains of GRK5 (see Figure 1) was proposed to be consistent with a common GRK activation mechanism mediated by lipids and GPCRs. The same sites did not however emerge with the use of a “zero-length” crosslinker and instead captured interactions between regions known to be highly dynamic in both proteins. HDX-MS results were largely consistent with the small lobe and lipid-binding regions becoming stabilized in the β2AR micelle/GRK5 assembly (although the peptide spanning the N-terminal lipid binding domain also includes several turns of the αN helix, complicating interpretation) [49]. Interestingly, the α5–α6 region of the GRK5 RH domain also exhibited decreased dynamics in the presence of activated β2AR. This would at face value seem to be in conflict with higher dynamics expected upon disruption of the RH-large lobe interface, but could be explained if this region in turn interacted with either the membrane or the receptor. Such would also be consistent with some of the crosslinking results. Conversely, only ICL3 of the β2AR could be shown to be stabilized in the presence of GRK5. Because prior studies suggest the involvement of multiple ICLs of the receptor [68], this result implies that other ICLs are similarly rigid in both GRK5-bound and free states. The results were also consistent with a prior HDX-MS study investigating the dynamics of bovine GRK1 in the presence or absence of light-activated rod outer segments, Mg^2+^·ATP, and the N-terminal αN helix [69], where it was found that GRK1·ATP in the presence of light activation of ROS exhibited a general decrease in dynamics not only in the RH domain—particularly, in its terminal subdomain—but also in the C-terminal end of αN, the AST region, and the P-loop of the active site. 

Based on these various lines of evidence, three illustrative models were generated to represent the conformational changes that occur in GRK5 during the process of activation by the β_2_AR [49]. Initially, GRK5 docks on the membrane in a compact conformation, with the RH bundle and large lobe of the kinase domain engaged with each other. Upon engagement with he β_2_AR, their interface is disrupted, leading to the release of the RH bundle subdomain from the large lobe of kinase domain. A positively charged patch on the RH domain α5 helix then interacts with acidic lipids on the membrane near ICL2, stabilizing the β_2_AR-GRK5 complex and leading to full activation of the kinase. In the model of this activated complex (Figure 2B), the N-terminal lipid-binding domain of GRK5 (residues 22–29) occupies the same cleft as the C-terminal helix of Gα and the finger loop of arrestin in their respective GPCR complexes.

However, there are several important issues left unresolved by this model. The first is that the N-terminal helix of GRK, which is known to play a critical role in receptor phosphorylation (Section 3.1.1), is positioned outside of the receptor interface, and no explanation was given for its role, although it could directly interact with phospholipids, as has been previously proposed [59]. Secondly, the GRK5 NLBD is known to activate GPCR phosphorylation via binding to negatively charged phospholipids, and thus, it seems unlikely that this region could bind to both receptor and anionic lipids at the same time due to steric exclusion. Furthermore, only the GRK4 subfamily has thus far been reported to have an NLBD, meaning that the other GRK subfamilies might require an entirely different docking site to recognize an activated receptor. Finally, the highly basic signature of the NLBD seems inconsistent with the chemical environment of the receptor cytoplasmic cleft, which binds to hydrophobic and acidic signatures in Gα subunits and arrestins. 

#### 3.1.3. The NLBD/RH Domain as the Primary GPCR Docking Site, but Instead Binding to ICL3 (Model 3)

GRK1 (rhodopsin kinase) and rhodopsin are unique among GRK-GPCR pairs, because they have coevolved since the emergence of vertebrates to regulate their perception of light. Thus, they represent a highly optimized system and a logical place to start in understanding GPCR–GRK interactions. A recent study used a cell-based proximity assay and in vitro binding assays, along with negative-stain EM and HDX-MS, to examine the interactions of GRK1 and GRK5 with rhodopsin [64]. For the cell-based assay, the authors used a modified Tango assay wherein a TEV proteolytic site and the tetracycline transactivator (tTA) are fused to the C-terminal tail of rhodopsin, and TEV protease is fused to the C-terminus of GRK1 (presumably eliminating its farnesylation site). When GRK1 and rhodopsin are in close proximity, proteolytic release and nuclear localization of tTA occurs, inducing the expression of a luciferase reporter as an indirect readout of the colocalization. The in vitro assay was a fluorescence-based AlphaScreen assay to directly measure either GRK1 binding to detergent solubilized rhodopsin or arrestin binding to GRK1-phosphorylated rhodopsin as an indirect readout for GRK1 activity. The Tango assays suggested that the RH domain of GRK1 constitutes the primary site for receptor binding, because the 1–183 (RH domain), 31–183 (RH domain lacking N-terminal helix and subsequent loop), and 31–563 (full length lacking N-terminal helix) truncations all showed the induction of luminescence at the same level as full-length GRK1. The AlphaScreen direct binding assay also showed no significant loss in signal when the N-terminus was deleted from any of the tested GRK1 variants. However, because there was no negative control reported that exhibited any significant reduction in the signal, it is not possible to draw definitive conclusions. On the other hand, although the N-terminal helix of GRK1 was deemed unnecessary for direct binding, the 31–563 truncation completely lost its ability to phosphorylate rhodopsin in both the AlphaScreen indirect kinase assay, as well as in a direct kinase assay, consistent with prior observations. The N-terminal mutants L6A and F15A also showed diminished luciferase induction in the Tango assay. Thus, the N-terminal helix was, regardless, deemed important for receptor interaction and/or kinase activation. Disruption of active site residues in GRK1 led to a loss of the Tango assay signal, but an addition of the ATP competitive inhibitor paroxetine did not. This was presented as evidence that the assembly was dependent on an active conformation of GRK1. However, it should be noted that paroxetine stabilizes inactive conformations of GRK1 [70] and GRK2 [71]. 

HDX-MS was then used to compare the dynamics of surfaces in human apo GRK1 versus a covalently fused rhodopsin-GRK1 protein, which was presumed to be in complex [64]. Overall, changes in the deuterium uptake were small (on the order of ±10% when observed), but most regions with significant changes registered an increase in uptake in the fusion protein, indicating more dynamic behavior overall, including peptides involved the RH bundle–large lobe interface. However, the activation status of rhodopsin and GRK1 in the fusion was not described, rendering the interpretation and comparison to prior rhodopsin–GRK1 HDX-MS studies [69] problematic. Mutagenesis studies followed by the Tango assay suggested hydrophobic residues in the loop between the N-terminal region and the first helix of the RH domain, as well as the α9 and α10 helices of the RH domain, were important for luciferase induction. Finally, negative-stain EM reconstruction of a rhodopsin-GRK5 fusion protein was interpreted to show the ICL3 loop of active rhodopsin interacting with the RH domain and the kinase domain interacting with the H8 region (or possibly vice versa because at the low resolution of the density the fit could be made either way) (Figure 2C).

There are also unresolved issues with this third model. For example, the human GRK1-R194A mutation, which obliterates enzyme activity in bovine GRK1 (R191A), as do the analogous mutations in GRK2 and GRK6 [61], did not perturb proximity in the Tango assay. The positive results obtained using paroxetine as a GRK “agonist” is also problematic [70], as is the fact that nearly all rhodopsin mutations tested in this study had no effect in the Tango assay, unlike in a prior study [68]. Finally, the proposed model would be difficult to extend to other GRK–GPCR complexes, in particular those with variable lengths of ICL3. The study did however reconfirm that the GRK N-terminus is important for receptor phosphorylation. To incorporate this idea, the authors proposed that the role of the N-terminal helix was only to stabilize the active kinase domain conformation when other regions were engaged with the receptor [36].

### 3.2. Extension of the Newer Models to the GRK2 Subfamily

In the two newer models (Section 3.1.2 and Section 3.1.3), a similar region of the GRK RH domain and/or its associated lipid-binding domains are proposed to be involved in direct GPCR binding, although with somewhat different regions of the receptor. Because the supporting studies for these models were members of the GRK1 and GRK4 subfamilies, which are closely related, it was unclear if these models could be extended to the more distantly related GRK2 subfamily. A key difference is that the RH domains of GRK2/3 do not play a direct role in phospholipid binding but, instead, scaffold a C-terminal PH domain that binds to both anionic phospholipids and Gβγ subunits, which are prenylated. Both of these interactions are required for GRK2 activity in cells. None of the three models compared in 3.1 directly speak to where Gβγ might be situated in the GPCR complex, but in Model 2, the data called for a potential reorientation of the RH domain as part of the activation mechanism. This would have a major impact on the position of Gβγ and the PH domain if it were also true in GRK2 and, perhaps, may require release of the PH domain from the RH domain in the GPCR complex. Thus, we felt it worth testing whether the analogous membrane proximal region of the GRK2 RH domain might also play a role in receptor binding.

When the kinase domain of GRK2 is superimposed onto those of GRK1/GRK5, the proposed receptor interacting regions in GRK1 and GRK5 are in proximity to the unique structures in GRK2 bearing hydrophobic residues that are solvent exposed in all reported crystal structures for the enzyme (Figure 3A). These residues are uniquely conserved in the GRK2 subfamily (Figure 3B) and, as such, could represent an unrecognized protein–protein interaction site in GRK2 and GRK3. Several acidic residues that are also uniquely conserved in the GRK2 subfamily are also part of this surface. Thus, if Models 2 and 3 are correct, we hypothesized that these GRK2 subfamily-specific conserved residues might interact with either the cytoplasmic cleft or ICL3 of the activated GPCRs. To test this hypothesis, we generated the L33N, E36A, E532A, L536N, and L547N variants in the background of C-terminally H_6_-tagged bovine GRK2 and purified them from the HEK293F cells. We then determined their steady-state kinetic parameters using either tubulin or rhodopsin in the rod outer segments (ROS) as substrates. The inactive K220R variant was generated as a negative control for background endogenous GRK activity that may copurifyfrom HEK293F cells. 

We hypothesized that there would be no change in the *K*_m_ values or normalized V_max_ for tubulin phosphorylation, because we did not expect these mutations would affect the binding kinetics of ATP or the ability of the kinase domain to adopt a closed state in the absence of a receptor. Similarly, we did not expect changes in the *K*_m_ for rhodopsin phosphorylation when ATP was varied, but changes in the normalized V_max_ for rhodopsin could be indicative of a receptor binding defect. However, we only observed small changes in the *K*_m_, normalized V_max_, or normalized V_max_/*K*_m_ (henceforth referred to as “normalized catalytic efficiency”) using either tubulin or rhodopsin (Figure 4 and Table 1 and Table 2). The biggest differences were in the normalized V_max_ data while varying rhodopsin concentration, but even so, the effects were less than two-fold.

A Michaelis–Menten analysis was also performed by varying the concentration of rhodopsin in ROS with a fixed concentration of ATP (Figure 5). Here, variants with receptor binding effects would be expected to increase the *K*_m_. However, there was no significant difference in the *K*_m_ for rhodopsin among any of the variants. We did observe statistically significant decreases in normalized V_max_ for all of the variants except L33N, but none of these variants exhibited significant differences in the normalized catalytic efficiency (Table 3). It is therefore likely that these differences could be due to differences in the specific activity towards rhodopsin among the purified variants. Notably, we previously prepared the E36A variant in the background of untagged GRK2 expressed in COS-1 lysates and, consistent with this work, the mutant retained >85% of WT phosphorylation activity when using 10-µM rhodopsin as a substrate [42].

To test these variants against a hormone responsive GPCR, we assessed their ability to phosphorylate the β_2_AR in COS-7 cells [52]. In a previous work, we found that at the isoproterenol (ISO) concentration commonly used to stimulate the β_2_AR in desensitization assays (10 µM) transfection of GRK2 only stimulated WT β_2_AR phosphorylation 40% [54]. In that work, we utilized the β_2_AR-Y326A variant, which is not phosphorylated by endogenous COS-7 cell kinases and therefore allowed the measurement of β_2_AR phosphorylation by transfected GRK2 [74,75]. For this work, we further adapted the assay to allow the measurement of WT β_2_AR phosphorylation by transfected GRK2. To optimize the agonist- and GRK2-dependent phosphorylation of WT β_2_AR, we measured its phosphorylation as a function of the ISO concentration in the absence and presence of transfected GRK2. The EC_50_ values for the phosphorylation of β_2_AR were 51 and 2.5 nM, respectively, in the absence and presence of transfected GRK2 (Figure 6A). The increase in ISO potency due to GRK2 was surprising but could be attributed either to an allosteric effect on the β_2_AR by the RH, kinase, or PH domains of GRK2 or to a phosphorylation event that directly or indirectly impacts the conformation of β_2_AR. Similar results have been observed for the μ-opioid receptor using arrestin recruitment assays as an indirect measure of GRK2 function. In cells stably or transiently transfected with the μ-opioid receptor, the transfection with GRK2 increased the potency of six agonists 10- to 44-fold (average of 28-fold) in β-arrestin1 (arrestin2) [76] or β-arrestin2 (arrestin3) [77] BRET recruitment assays. However, the CRISPR knockout of GRK2 in HEK293 cells diminished the maximum signal, but not the potency of agonist-induced recruitment to the μ-opioid receptor in a β-arrestin1 (arrestin2) BRET recruitment assay [78]. In combination, these results suggest that the potency-increasing effects of GRK2 might occur only in tissues such as the brain where the GRK2 expression level is high. The level of GRK2 in the rat cerebellum and cortex is similar to the level found in transfected cells [65].

In the dose-response assay, 2-nM ISO promoted β_2_AR phosphorylation >20-fold more in the presence of GRK2 than in its absence. We therefore tested the ability of WT GRK2-H_6_ and the L33N, E36A, E532A, L536N, and L547N variants to phosphorylate WT β_2_AR using 2-nM ISO to stimulate the receptor. β_2_AR phosphorylation was abolished when alprenolol, an inverse agonist, replaced ISO or when the kinase-deficient K220R was used in the assay (Figure 6B). In contrast, β_2_AR phosphorylation by the RH domain mutants were comparable to WT GRK2. These data were therefore consistent with the data derived from in vitro kinetic assays (Figure 4 and Figure 5), suggesting that this membrane proximal region of GRK2 does not play an important role in GPCR interactions, at least under the conditions we tested. The possibility remains that more disruptive mutations than the point mutants tested here, such as the introduction of double or triple mutants, would be required to observe a receptor-binding defect.

## 4. Discussion

In Model 1, which was based largely on structural and functional studies and comparison with other GPCR–protein complexes, the N-terminus of GRKs is proposed to serve as the key GPCR-binding element and to dock directly in the cytoplasmic cleft of the activated GPCR. Although there is no structural evidence yet reported for this model, the N-terminal helix interactions with the small lobe, and AST would clearly help stabilize a more closed, active conformation of the kinase domain. This provides a ready mechanism for how GRKs become activated if GPCR binding facilitates the folding and packing of the N-terminal helix. This model could also be applied uniformly across all GRKs, because, unlike the NLBD, the N-terminus is highly conserved among GRKs. Furthermore, this receptor docking model would accommodate different lengths of ICL3 in GPCRs.

In contrast, the newer Models 2 and 3 suggest no direct role for the N-terminus in receptor binding. Instead, the supporting data pointed to docking sites in regions within or immediately adjacent to the terminal subdomain of the RH domain. Since this region varies considerably in sequence among the GRK subfamilies, these models spurred us on to test whether the analogous region in the RH domain of GRK2 could play an important role in GPCR binding. However, our data indicated that the GRK2 RH domain and residues in its preceding loop do not play a major role in GPCR binding, regardless of whether the receptor has a short (rhodopsin) or longer ICL3 (β_2_AR), and regardless of whether one measures phosphorylation with purified components in vitro or in living cells. Point mutations of residues in the N-terminus of GRK2, in contrast, have a much more profound impact on GPCR phosphorylation in living cells and in vitro [54]. Another issue with Model 2 is that it suggests GRK5 activation proceeds largely through disruption of an interface between the RH domain and the large lobe, even though this interface is small, highly variable among available crystal structures, and mutations in the interface do not exhibit an activating effect in either GRK1 [64] or GRK2 [42].

Is there a way to reconcile the three GPCR docking models and the new GRK2 data reported here? One simple explanation would be that individual GRK subfamilies interact with individual GPCRs using distinct mechanisms. This, however, seems unsatisfying given their close evolutionary relationship (particularly between the GRK1 and GRK5 subfamilies) and the fact that individual receptors, with poor conservation in their ICL regions, can typically be phosphorylated efficiently by multiple GRKs. A more universal mechanism for GPCR engagement by GRKs therefore seems warranted. Differences in the models could also stem from the confounding requirement of GRKs for the presence of both anionic phospholipids and the activated receptor conformation. In other words, the loss of signal being measured in some of the supporting experiments may simply reflect the propensity of the GRK being studied (GRK5 and GRK1) to bind to membranes (or bicelles) rather than the receptor itself. For example, GRK5 binds to negatively charged membranes regardless of the presence of an agonist-bound receptor, as was the case in the studies backing Model 2, where it was pulled down in the presence of an inverse agonist. It should also be noted that GRK4 subfamily members have a greater propensity to phosphorylate inactive GPCRs in intact cells [79]. The Tango assays used to support Model 3 will also exhibit diminished signal if GRK1 could not be efficiently recruited by a lipid binding domain. An anionic lipid binding domain in GRK1 has not yet been reported, but, as a relatively close homolog of GRK5, it retains a strong basic signature in the region analogous to the NLBD. In other words, although receptor binding is expected to be transient, lipid binding can be persistent, especially in the case of GRK5. Thus, some of the conflicting data could be explained by mutations that interfere with the ability of the GRK to bind membranes or anionic lipids. By analogy, it is not trivial to know whether arrestin is binding to the core of an activated receptor, its phosphorylated tail, or both at the same time. How GPCRs interact with and drive the activation of GRKs will, however, remain an open question until a high-resolution structure of a confirmed agonist-dependent GRK–GPCR complex is elucidated and experimentally validated.

## Figures and Tables

**Figure 1 biomolecules-11-00447-f001:**
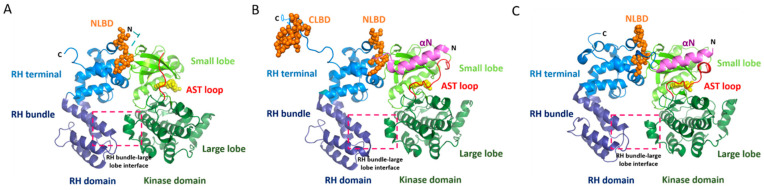
Crystal structures representing inactive and active conformations of the G protein-coupled receptors kinase 4 (GRK4) subfamily members. (**A**) Crystal structure of GRK6 in a complex with adenylyl imidodiphosphate (Protein Data Bank (PDB) entry: 2ACX). (**B**) Crystal structure of GRK6 in a complex with sangivamycin (PDB: 3NYN). (**C**) Crystal structure of GRK5 in complex with Ca^2+^·CaM (PDB: 6PJX), with Ca^2+^·CaM removed for the sake of comparison. The regulator of the G protein signaling homology (RH) and kinase domains are shown in blue and green, respectively. The active site tether (AST) loop and N-terminal (αN) helix (disordered in panel **A**) are highlighted in red and magenta, respectively. The N-terminal lipid-binding domain (NLBD) and C-terminal lipid-binding domain (CLBD) are shown with orange spheres in the structures where they were ordered. The RH bundle–large lobe interface is highlighted with a red dashed box. Ligands bound in the active site are shown with yellow spheres. Note the relative closure of the kinase domain in panels **B** and **C** relative to panel **A**, as well as the ordering of the αN helix and the AST region, which coalesce in the activated structures.

**Figure 2 biomolecules-11-00447-f002:**
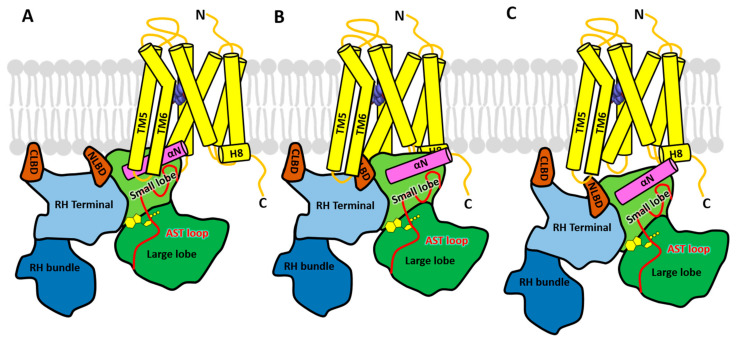
Three models of the GPCR–GRK complex. GRK5 is modeled in complex with the β_2_-adrenergic receptor (yellow cylinders) bound to the agonist epinephrine (purple spheres). The coloring of the structural domains of GRKs is the same as in Figure 1. (**A**) Model 1, in which the N-terminal helix (αN) forms the primary docking site, was conceived based on crystal structures of GRKs and functional studies of mutants in each of the three GRK subfamilies, and modeled based on similarities with known heterotrimeric G protein complexes with receptors. (**B**) Model 2, in which the NLBD instead docks in the cytoplasmic cleft, is derived primarily from crosslinking with mass spectrometry (CLMS), hydrogen-deuterium exchange mass spectrometry (HDX-MS), and computational studies of a β_2_-adrenergic receptor (β_2_AR) assembly with GRK5 [49]. The αN helix is not directly involved in receptor contacts. (**C**) Model 3, wherein the NLBD and terminal subdomain are modeled to engage ICL3, was created based primarily on cell-based assays measuring rhodopsin and GRK1 proximity, as well as negative stain electron microscopy (EM) studies of a rhodopsin-GRK5 fusion protein [64]. Unlike the first two models, no element occupies the receptor cytoplasmic cleft. Like Model 2, the αN helix is not directly involved in receptor interactions but proposed to still be important for stabilizing an active kinase domain configuration.

**Figure 3 biomolecules-11-00447-f003:**
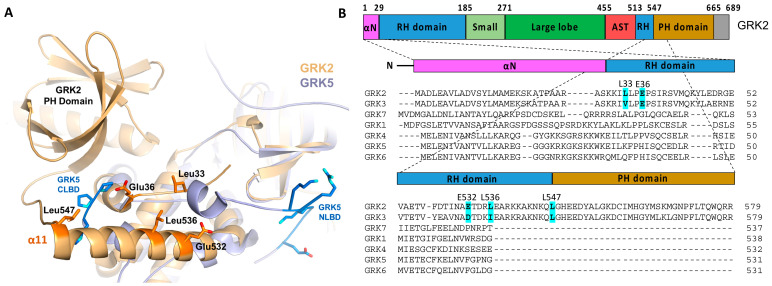
Hypothetical GPCR-interacting sites in the GRK2 RH domain. (**A**) GRK2 (PDB: 4PNK [72], gold) aligned with full-length GRK5 (PDB: 4TND [73], slate) indicated the presence of a patch of solvent-exposed residues uniquely conserved in GRK2/3 and similarly disposed with respect to the membrane as the NLBD and CLBD of GRK5. These residues could conceivably interact with activated GPCRs [49]. Side chains of the GRK2 residues that were modified in this study are depicted as orange sticks. (**B**) Domain diagram of GRK2 and the structure-based sequence alignment of the seven human GRKs, with the targeted positions in GRK2 highlighted.

**Figure 4 biomolecules-11-00447-f004:**
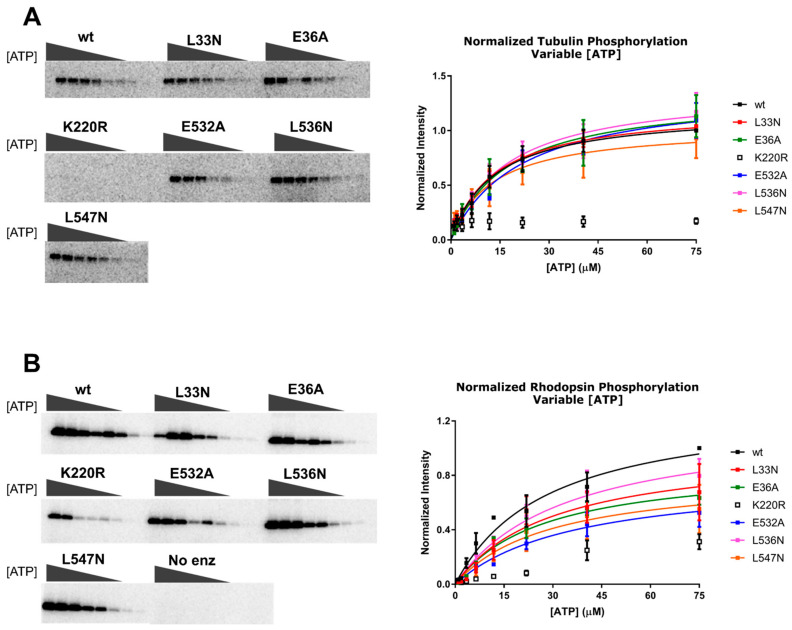
Michaelis–Menten analysis of the GRK2 variants varying ATP concentration ([ATP]). Bovine GRK2 variants with a C-terminal H_6_ tag purified from HEK293F cells were analyzed by Michaelis–Menten kinetics. (**A**) Tubulin and (**B**) rod outer segments (ROS) phosphorylation by the GRK2 mutants, as detected by SDS-PAGE (left, corresponding to the results from a single representative experiment). Bands correspond to the phosphorylated product at variable [ATP] (0–75 µM). Band intensities were fitted to the Michaelis–Menten model using GraphPad Prism (right). Plots contain data from three experiments, and error bars represent the standard deviation (SD).

**Figure 5 biomolecules-11-00447-f005:**
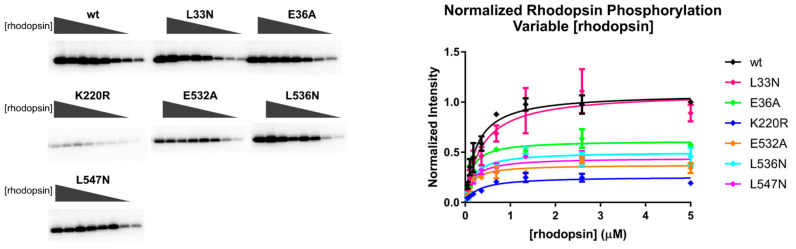
Michaelis–Menten analysis of the GRK2 variants varying the concentration of rhodopsin [rhodopsin]. Phosphorylation of ROS by GRK2 mutants was assessed by SDS-PAGE (left, single representative experiment), and band intensities corresponding to the phosphorylated product from experiments performed in triplicate were plotted as a function of [ROS] (right). Error bars represent the SD.

**Figure 6 biomolecules-11-00447-f006:**
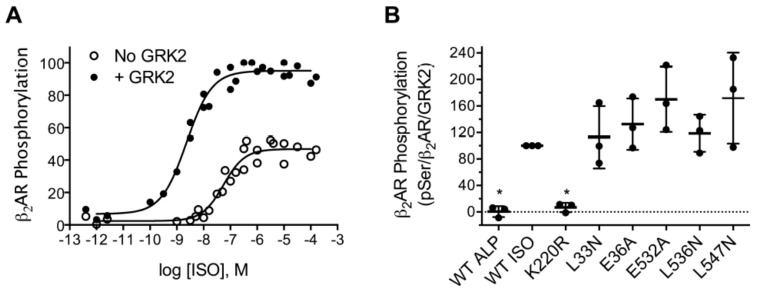
Intact cell phosphorylation of WT β_2_AR. COS-7 cells were transfected with FLAG-tagged WT β_2_AR, Gα_s_, Gβ, Gγ, and GRK2 (**A**) or GRK2-H_6_ and its RH domain mutant variants (**B**), where indicated. Cells were stimulated with 2-nM isoproterenol (ISO), and lysates were subjected to immunoblotting. Phosphorylated β_2_AR was detected with a pSer555/pSer556 phosphorylation site antibody (pSer), total β_2_AR was detected with an antibody that recognizes the carboxyl tail of the receptor (β2AR), and GRK2 was detected with a polyclonal antibody. (**A**) Phosphorylated β_2_AR as a function of the ISO concentration (dose response). Shown are all the data points from three independent experiments. (**B**) Phosphorylation by GRK2 variants. The level of phosphorylation was normalized to the levels of both the total receptor and GRK2 and expressed as a percentage of the WT. Data from three independent experiments are shown. No statistically significant differences between WT and RH domain mutants were observed. * *p* < 0.05. Error bars show the mean and SD.

**Table 1 biomolecules-11-00447-t001:** Steady-state parameters for tubulin phosphorylation when [ATP] is varied.

Bovine GRK2 Variant	*K*_m_ (95% CI) (µM)	Normalized V_max_ (95% CI) (A.U.) ^1^	Normalized V_max_/*K*_m_ (µM^−1^)
WT	13 (8 to 21)	1.2 (1.0 to 1.4)	0.09
L33N	13 (10 to 18)	1.2 (1.1 to 1.3)	0.09
E36A	17 (8 to 44)	1.3 (1.0 to 1.9)	0.08
K220R	N.D. ^2^	N.D.	N.D.
E532A	22 (12 to 40)	1.4 (1.1 to 1.8)	0.06
L536N	17 (9 to 32)	1.4 (1.1 to 1.8)	0.08
L547N	13 (4 to 36)	1.0 (0.8 to 1.6)	0.08

^1^ A.U., arbitrary units and ^2^ N.D., not determined. The K220R tubulin phosphorylation signal was too low to fit.

**Table 2 biomolecules-11-00447-t002:** Steady-state parameters for rhodopsin phosphorylation when [ATP] is varied.

Bovine GRK2 Variant	*K*_m_ (95% CI) (µM)	Normalized V_max_ (95% CI) (A.U.) ^1^	Normalized V_max_/*K*_m_ (µM^−1^)
WT	24 (16 to 36)	1.3 (1.1 to 1.6)	0.05
L33N	30 (14 to 82)	1.0 (0.7 to 1.7)	0.03
E36A	27 (18 to 43)	0.9 (0.8 to 1.1)	0.03
K220R	N.D. ^2^	N.D.	N.D.
E532A	37 (17 to 101)	0.8 (0.6 to 1.4)	0.02
L536N	34 (17 to 77)	1.2 (0.9 to 1.8)	0.04
L547N	24 (9 to 89)	0.8 (0.5 to 1.5)	0.03

^1^ A.U., arbitrary units and ^2^ N.D., not determined. The K220R rhodopsin phosphorylation signal was too low to fit.

**Table 3 biomolecules-11-00447-t003:** Steady-state parameters for phosphorylation when [rhodopsin] is varied.

Bovine GRK2 Variant	*K*_m_ (95% CI) (nM) ^1^	Normalized V_max_ (95% CI) (A.U.) ^2^	Normalized V_max_/*K*_m_ (µM^−1^)
wt	240 (160 to 350)	1.1 (1.0 to 1.2)	5
L33N	360 (160 to 760)	1.1 (0.9 to 1.4)	3
E36A	130 (75 to 220)	0.6 (0.5 to 0.7)	5
K220R	300 (160 to 510)	0.3 (0.2 to 0.3)	1
E532A	140 (70 to 260)	0.4 (0.3 to 0.4)	3
L536N	180 (90 to 330)	0.5 (0.4 to 0.6)	3
L547N	170 (93 to 290)	0.4 (0.4 to 0.5)	2

^1^ Note that the *K*_m_ values for variable [rhodopsin] are reported in nM, while the *K*_m_ values for variable [ATP] (Table 1 and Table 2) are reported in µM; ^2^ A.U., arbitrary units.

## Data Availability

The data presented in this study are available on request from the corresponding author.

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
