# Peer review of "The Open Question of How GPCRs Interact with GPCR Kinases (GRKs)"

_biomolecules, 2021, doi:10.3390/biom11030447_

Round 1
Reviewer 1 Report
Cato and colleagues have produced a comprehensive and highly-readable review article focussed on GRK interactions with GPCRs, addressing specifically three competing models for this interaction. This review is important, as the authors are experts in the field and this is the only major GPCR-transducer complex for which direct structural data do not yet exist. The review is particularly strong in that it clearly lays out the strengths and weaknesses of the various approaches to the question, as well as the caveats associated with interpretation of each approach. The authors apply the same critical eye to the model that they clearly favor as to the competing models. Overall an excellent piece of scholarship that is very informative.
My only criticism of the manuscript relates to the experimental section appended to the review, which is meant as a test of the competing interaction models. While well-conceived and interesting, adding these data changes the manuscript to something that's neither a review or quite a research report. As a research report, while the data shown in Figure 4B don't clearly indicate a binding defect, neither do they convincingly rule it out. The Vmax values are fairly scattered, and this only appears to be a single experiment done in triplicate. Under the circumstances it's not clear what "no significant changes" (line 377) really means, or how confident one can be that this experiment rigorously tested the hypothesis.
Author Response
We agree that the inclusion of the experimental data section was unorthodox. However, we felt it was appropriate because the test was conceived based on the review. We have now moved the analysis portion into the results at the suggestion of another reviewer, which makes the paper more conventional. We have altered the text describing 4B to say that while the Vmax values are more scattered, which makes it difficult to rule out that there is no Vmax effect, the catalytic efficiencies are similar and there is nothing deficient enough to be consistent with a receptor binding site. Point mutations in other regions (N-terminal helix and AST) are on the other hand cause more dramatic effects.
The experiment shown is an n=3 from a single preparation, run on different days with different dilutions. The gel is representative of one of those experiments, but the plot contains data from three different experiments.
The bottom line here is that we were not seeing dramatic effects based on the sum total of the data (<2-fold effects), and we concluded that further attempts to reduce the spread of the data in this case are not going to be worthwhile.
Reviewer 2 Report
This very interesting review discusses three distinct models for how GRKs bind and recognize activated GPCRs. They critical discuss the studies and data supporting the different models and also point out limitations in the approaches and additional experiments that should be performed. One of these experiments, testing of an RH Domain GPCR Docking Site in GRK2, they performed themselves. For this they generated 6 variations each containing a single mutation within the GRK RH domain and subsequently tested them for binding and activity. The data clearly show that none of the single mutations affects binding or activity, making model 2 and 3 less likely. However, as indicated by themselves in the discussion, they cannot exclude that a combination (double, triple, etc.) is necessary to completely disrupt binding. Since I greatly appreciate that the authors included primary data in this review, I would not ask them to perform additional experiments. Instead I suggest to move the sentence discussing these limitation (line 481 – 483) to the results part (after line 446), to make this limitation in the approach more clear.
Author Response
Thank you for your positive review and we have moved the statement as indicated.
Reviewer 3 Report
This is a nice review highlighting the current status of GPCR-GRK interaction. I have only a few minor issues:
Authors could discuss how the association between PH domain of GRK2 family with Gbg subunit may fit into these models.
Additional information could also clarify whether the GRK-lipid binding occurs prior GPCR binding for GRKs, how it may be integrated into these three models.
The fonts of some text are different from the rest of text.
Author Response
This is a nice review highlighting the current status of GPCR-GRK interaction. I have only a few minor issues:
Thank you for the positive feedback.
Authors could discuss how the association between PH domain of GRK2 family with Gbg subunit may fit into these models.
This is a good suggestion, and consequently we added some text at the beginning of the GRK2 experimental section (because we are trying to extend the key takeaway from these models to GRK2) to have a brief discussion with respect to Gbg regulation of GRK2/3 enzymes. In a nutshell, none of the models here speak to where Gbg might be in the complex or it might be doing. However, in Model 2, where flexibility of the RH domain is suggested as part of the activation mechanism, this could have important ramifications for regulation by Gbg in the GRK2/3 subfamily because the RH domain scaffolds the PH domain.
Additional information could also clarify whether the GRK-lipid binding occurs prior GPCR binding for GRKs, how it may be integrated into these three models.
It is clear that GRKs can bind to membranes in the absence of activated receptors, in part because of observations that peptide phosphorylation can be enhanced just by lipids, and in part because GRK5 is membrane bound already in cells. So it would be reasonable to propose that GRKs bind to lipids first. But at what stage any sort of allosteric activation conferred by lipids are imposed upon GRKs during phosphorylation of GPCRs has not been studied, and thus we feel we cannot comment. It is our believe that both need to happen simultaneously, regardless of order, because in the absence of either, one loses receptor phosphorylation.
The fonts of some text are different from the rest of text.
We will fix this in our revision. Thank you for pointing it out.
Reviewer 4 Report
This is an interesting and deep analysis of the current hypotheses for GPCR-GRK interactions, coming from the experts in the field. The authors propose 3 illustrative models and discuss them in the context of several biochemical and structural data. They also test whether RH domain of GRK2 interacts with GPCRs, which would corroborate models 2 and 3. The mutations introduced in RH domain, however, did not change phosphorylation, “confirming that this membrane proximal region of GRK2 does not play an important role in GPCR interactions, at least under the conditions we tested. This is a useful negative result that could be worthy of publication.
The manuscript, however, has a confusing structure, where the first half reads like a review and the second as an original article. To conform to either type of publication, it needs some rearrangements. If the models in Figure 2 are new, they should be included as “hypothetical interaction models” as a part of “Results” section that starts after the Introduction. The “Conclusion” also reads more like a short discussion and should be named correspondingly.
Several specific issues should also be addressed:
Lines 230-255: this paragraph discussing several EM, crosslinking and HDX results goes without a single reference. Is it an omission, or are these results original unpublished data from the authors? Please clarify.
Lined 265-275: “three computational models of β2AR-GRK5 complex were generated”
How these 3 models correspond to the 3 models shown in Figure 2? Or are they different “stages” of Model 2 discussed in this section?
Also, there is no reference for the “three computational models”. There is some wording describing the hypothesis, but they do not constitute a “computational model” unless there is some real structure-based computational modeling with results presented as a PDB file. Please clarify and either (a) change the wording to avoid “ computational model” (e.g. use hypothesis, potential arrangement, or “illustrative model”)
In several other statements that lack citation, it is not obvious whether new results are reported or the previous studies referenced. This should be made clear.
Author Response
This is an interesting and deep analysis of the current hypotheses for GPCR-GRK interactions, coming from the experts in the field. The authors propose 3 illustrative models and discuss them in the context of several biochemical and structural data. They also test whether RH domain of GRK2 interacts with GPCRs, which would corroborate models 2 and 3. The mutations introduced in RH domain, however, did not change phosphorylation, “confirming that this membrane proximal region of GRK2 does not play an important role in GPCR interactions, at least under the conditions we tested. This is a useful negative result that could be worthy of publication.
The manuscript, however, has a confusing structure, where the first half reads like a review and the second as an original article. To conform to either type of publication, it needs some rearrangements. If the models in Figure 2 are new, they should be included as “hypothetical interaction models” as a part of “Results” section that starts after the Introduction. The “Conclusion” also reads more like a short discussion and should be named correspondingly.
We have now reformatted the paper as a research article. The models shown are not “new”, in that they were proposed by us and other groups before. Therefore we have moved this section into the Results as “Analysis of GPCR-GRK interaction models” and to give the paper more of an overall feel as an “article”.
Several specific issues should also be addressed:
Lines 230-255: this paragraph discussing several EM, crosslinking and HDX results goes without a single reference. Is it an omission, or are these results original unpublished data from the authors? Please clarify.
In the revision we (hope we) have clarified in this section that these experiments were from a single source paper, referenced at the beginning of those sections. We have altered the structure of these sections to make it more clear.
Lined 265-275: “three computational models of β2AR-GRK5 complex were generated”
How these 3 models correspond to the 3 models shown in Figure 2?
They are models of the progression from the inactive state to the GPCR engaged state of Model 2 based primarily on their crosslinking and MD simulations.
Or are they different “stages” of Model 2 discussed in this section?
Correct.
Also, there is no reference for the “three computational models”.
It is from the same paper referenced at the beginning of the section. We have clarified this in the revision.
There is some wording describing the hypothesis, but they do not constitute a “computational model” unless there is some real structure-based computational modeling with results presented as a PDB file. Please clarify and either (a) change the wording to avoid “ computational model” (e.g. use hypothesis, potential arrangement, or “illustrative model”)
We decided to go with “illustrative”.
In several other statements that lack citation, it is not obvious whether new results are reported or the previous studies referenced. This should be made clear.
We will do so. Thank you for alerting us to the confusion about where data was coming from. We will re-reference the primary experimental paper where the bulk of these experiments are from and state at the outset what kinds of experiments are in each of the lead references for 3.1.2 and 3.1.3.
Round 2
Reviewer 4 Report
The authors adequately responded to the reviewer's suggestions, substantially improving the paper. I am OK with accepting for publication.